# Wi-Fi Assisted Contextual Multi-Armed Bandit for Neighbor Discovery and Selection in Millimeter Wave Device to Device Communications

**DOI:** 10.3390/s21082835

**Published:** 2021-04-17

**Authors:** Sherief Hashima, Kohei Hatano, Hany Kasban, Ehab Mahmoud Mohamed

**Affiliations:** 1RIKEN-Advanced Intelligent Project, Computational Learning Theory Team, Fukuoka 819-0395, Japan; hatano@inf.kyushu-u.ac.jp; 2Engineering and Scientific Equipment’s Department, Egyptian Atomic Energy Authority, Cairo 13759, Egypt; hany_kasban@yahoo.com; 3Faculty of Arts and Science, Kyushu University, Fukuoka 819-0395, Japan; 4Electrical Engineering Department, College of Engineering, Prince Sattam Bin Abdulaziz University, Wadi Addwasir 11991, Saudi Arabia; 5Electrical Engineering Department, Faculty of Engineering, Aswan University, Aswan 81542, Egypt

**Keywords:** millimeter-wave, machine learning, multi-armed bandit (MAB), contextual MAB, NDS, EA-LinUCB, EA-CTS

## Abstract

The unique features of millimeter waves (mmWaves) motivate its leveraging to future, beyond-fifth-generation/sixth-generation (B5G/6G)-based device-to-device (D2D) communications. However, the neighborhood discovery and selection (NDS) problem still needs intelligent solutions due to the trade-off of investigating adjacent devices for the optimum device choice against the crucial beamform training (BT) overhead. In this paper, by making use of multiband (μW/mmWave) standard devices, the mmWave NDS problem is addressed using machine-learning-based contextual multi-armed bandit (CMAB) algorithms. This is done by leveraging the context information of Wi-Fi signal characteristics, i.e., received signal strength (RSS), mean, and variance, to further improve the NDS method. In this setup, the transmitting device acts as the player, the arms are the candidate mmWave D2D links between that device and its neighbors, while the reward is the average throughput. We examine the NDS’s primary trade-off and the impacts of the contextual information on the total performance. Furthermore, modified energy-aware linear upper confidence bound (EA-LinUCB) and contextual Thomson sampling (EA-CTS) algorithms are proposed to handle the problem through reflecting the nearby devices’ withstanding battery levels, which simulate real scenarios. Simulation results ensure the superior efficiency of the proposed algorithms over the single band (mmWave) energy-aware noncontextual MAB algorithms (EA-UCB and EA-TS) and traditional schemes regarding energy efficiency and average throughput with a reasonable convergence rate.

## 1. Introduction

The drastically exponential growth of wireless traffic sparks future communication standards (beyond fifth generation, B5G, and sixth generation, 6G) to shift their operating bands from the crowdy sub 6 GHz band into the abandoned millimeter wave (mmWave), i.e., 30–300 GHz, band. Although mmWave has excellent positives such as huge available spectrum, large capacity, and the ability to support high data rates and bandwidth-intensive applications, it suffers from several negatives that represent the main obstacle to deal with. Millimeter-wave signals experience harsh path loss, blockage sensitivity, and absorption from the wireless environment due to their short wavelengths [1]. Consequently, directional communication usage by employing high gain antennas and beamforming training (BT) is advocated to overcome the significant attenuation at a considerable overhead expense. The short-range transmission of a mmWave enables device-to-device (D2D) communication, making it a hopeful future B5G/6G policy via relaxing the large traffic load on the cellular networks [2]. The small-distance D2D communication is well appropriate for the low-coverage mmWave transmission, while the out-band huge data rates given by the mmWave links can be beneficial for D2D. However, efficient and reliable mmWave D2D network constructions suffer from several fundamental problems, including the neighbor discovery and selection (NDS) [3,4,5]. Typically, direct mmWave NDS is used in mmWave D2D communications [6]—where a device first explores its adjacent devices by performing BT with them all, then selects the most suitable one for establishing the D2D linkage. Accordingly, this will consume a significant overhead, affecting the mmWave D2D networks’ throughput and energy consumptions. Besides, it neglects the adjacent devices’ remaining energies, which are generally crucial to apply D2D communication. That is, the selected device may not have sufficient energy for establishing the mmWave D2D link. However, direct NDS should be recurrently made to keep updating the environmental change like shadowing and instantaneous path blockage, which further hardens the problem.

Machine learning (ML) is a remarkable approach to deal with inevitable mmWave difficulties through its self-learning capability and effective decision-making [7,8]. Accordingly, this will mitigate the challenges of repeatedly investigating the surroundings by BT usage. Reinforcement learning (RL) is a vital ML branch, where the player explores the surroundings and tries to maximize its long-term rewards with no prior information about the environment. Hence, RL techniques are more promising solutions for mmWave communication systems than other ML methods like deep learning (DL). In DL, the learning process consumes a considerable amount of data, energy, time, and repetition times according to the changes in the scenario [9]. RL’s key challenge is the trade-off between holding the current choice and learning novel ones, officially recognized as the exploitation-exploration dilemma. Multi-armed bandits (MABs), firstly suggested by Auer [10], can efficiently deal with such trade-off. In MAB, a player interacts with several slot machines (arms) to increase her accumulated award. There are several MAB techniques; specifically, a valuable version is the contextual MAB (CMAB) type [11]. In CMABs, at every round *t*, an agent selects only one action from *K* ones, stated as *K* arms. Before deciding the arm to take, the agent looks at *d*-dimensional feature vectors, named “context”, related to each arm *k*. The agent utilizes this context information besides the arms’ pre-obtained rewards to decide the current round playing arm.

Academic and industrial researchers extensively examined the integration between the Wi-Fi and mmWave bands via exploiting their related properties to overcome the mmWave communication difficulties. Hence, IEEE 802.11 ad, ay standard [12] adapted the first dual-band (Wi-Fi /mmWave) technology. Different technology companies like QUALCOMM [13], Intel [14], and TP-Link [15] have created 2.4/5/60 GHz tri-band Wi-Fi demo chipsets products. Moreover, academically, many related research works investigated Wi-Fi/mmWave integrations [3,16,17,18,19]. In this paper, the mmWave D2D NDS is devised using CMAB schemes where the arms are mmWave D2D links constructed by the nearby devices. The reward, represented by the throughput of each mmWave D2D link, is drawn independently according to its line-of-sight (LOS) blockage probability. The context of each D2D linkage is its available Wi-Fi signal information, such as the received signal strength (RSS), mean, and variance of the Wi-Fi signal.

The motivation behind taking Wi-Fi information as a context of the mmWave D2D links is its ease of accessibility in multiband devices [3,16,17], which does not require extra processes. Furthermore, the authors in [3,16,17,18,19] proved the direct relationship between Wi-Fi and mmWave link statistics for multiband devices. The Wi-Fi signal statistics can predict mmWave link blockage’s probability and the likelihood of the mmWave RSS [3,16,17,18,19]. The main target of the modeled CMAB problem is to optimize the average throughput while considering the devices’ remaining energy. Therefore, energy-aware linear upper confidence bound (EA-LinUCB) and contextual Thomson sampling (EA-CTS) EA-CMAB algorithms are proposed by appending the remaining energy constraints to the original LinUCB [20] and CTS [21] CMAB algorithms. In the proposed EA-CMAB algorithms, the devices having residual energies above a specified limit will play the game, and the full game is finished when the whole devices reach the energy limit. Numerical investigations verify the outstanding performance of the EA-CMAB-based mmWave D2D NDS over both noncontextual EA-MAB proposed in [4] and traditional techniques. To the best of our knowledge, the current work is the first that proposes a ML-based context-aware bandit algorithm for mmWave D2D NDS.

The key contributions of this paper are highlighted as follows. Motivated by the standardized multiband devices, the mmWave D2D NDS optimization problem is modeled as budget constrained CMAB. The central device is the player, the arms are the nearby devices, the budget is the adjacent devices’ residual energies for constructing the D2D linkages, and the reward is the obtained throughput from the selected nearby device. Finally, the context is the nearby devices’ (arms) Wi-Fi information. We named the algorithms as EA-CMAB.

We examine the effect of Wi-Fi contextual information on the overall system performance by leveraging LinUCB [20] and CTS [21] algorithms and compare them with their noncontextual versions, i.e., UCB [22] and TS [23].

We propose EA-CMAB algorithms, e.g., EA-LinUCB and EA-CTS, for addressing the problem. In the proposed algorithms, the adjacent devices’ lasting energies are considered while performing online learning for selecting the best device for creating the mmWave D2D link.

Widespread numerical investigations are done to evaluate the proposed EA-CMAB-based algorithms at diverse situations and to examine their performances against two standard schemes named conventional direct NDS and random selection. Moreover, the proposed algorithms are compared with the noncontextual EA-MAB (EA-UCB and EA-TS) ones presented in [4].

The remainder of this paper is organized as follows. Section 2 reviews the related works. Section 3 introduces the mmWave D2D system model plus the utilized Wi-Fi and mmWave linkage models besides mmWave D2D NDS problem formulation, and the general concept of the CMAB algorithms. Section 4 discusses the proposed EA-CMAB algorithms. Section 5 gives the numerical investigations followed by the concluded remarks in Section 6. The following table shows the nomenclature used throughout this paper.

## 2. Literature Review

MmWave D2D communications carry great hopes to afford the capacity and the spectrum efficiency requirements of B5G/6G systems. A comprehensive study on mmWave and D2D related aspects like NDS, interference management, and network security are provided in [2,5,24], respectively. Furthermore, a comprehensive survey on D2D device discovery is provided in [25]. Designing an efficient NDS algorithm for mmWave D2D networks is more challenging due to high gain directional antenna usage and BT overhead. In [26], a novel D2D neighbor discovery algorithm that practices necklaces’ idea to mitigate the worst-case discovery latency compared with former methods is presented. Specifically, they leveraged Po’lya’s enumeration theorem and Fredricksen, Kessler and Maiorana (FKM) algorithm to discover briefer and effective scanning sequences for the nodes. However, the paper focused on the delay time only and neglected to maximize the accumulated reward.

A novel distributed algorithm using stochastic geometry tools that enable the devices to choose between the mmWave and μW bands for transmitting data by discovering unblocked mmWave LOS links was proposed in [27]. However, our proposed ML-based algorithms depend on mmWave band communications with side information from Wi-Fi and utilize mmWave for the whole data communications, not like [27] that switches between Wi-Fi and mmWaves. A novel cross-technology communication-based technique for neighbor discovery called NewBee that made use of coordination of Wi-Fi nodes to help neighbor discovery (ND) of Zigbee nodes is suggested in [28]. However, the authors did not consider mmWaves nor ML solutions in their proposal. In [29], a compressed-sensing related FastND algorithm that speeds up the ND process by dynamically learning the spatial channel characteristics is discussed. Although the authors made a successful practical experimental setup, their algorithm still does direct NDS with nearby devices and does not choose the best nearby device as in our case. In [30], the authors proposed a clustering scheme that splits the network nodes into clusters. Each cluster assigns one separate control channel and a particular mmWave channel for beamforming only. In [31], the authors suggested exploiting the context info associated with user position, handled by a separate control channel to advance the cell discovery process with minimizing its time delay. The schemes in [30] and [31] need an extra control channel, which increases the ND overhead, unlike our proposal that does not require any extra control channel. Employing linear programming, the authors of [32] proposed a distributed random mmWave-based discovery algorithm, where each device finds the relevant algorithm parameters, i.e., transmission and beam steering probabilities, using the information provided from the microwave band. However, they did not consider best neighbor selection besides the high complexity of linear programming especially for numerous adjacent devices. Another context-aware approach is provided in [33], where new cell discovery supported by the context information obtained from geo-located databases in heterogeneous mmWave networks was proposed. However, they did not consider mmWave D2D scenario, plus their method requires access to a previously established database, which might not be updatable, plus the labor work needed for constructing this database. A hunting-based directional neighbor discovery (HDND) technique for mmWave-based ad hoc networks is presented in [34]. However, it does not consider the D2D scenario nor applies advanced ML techniques.

Recently, MABs attracted significant attention in numerous sequential decision-making-based applications, especially in wireless networks [5,35,36,37,38]. In [5], we surveyed the applications of ML algorithms in different D2D communication challenges including NDS, resource allocation, power control, etc. To confirm the efficiency of ML in addressing these problems, we presented a case study of applying UCB and minimax optimal stochastic strategy (MOSS) algorithms in mmWave NDS problem. However, both applied algorithms were neither contextual nor energy aware ones. The authors of [4] leveraged stochastic bandit algorithms to solve similar problem by accounting the nearby devices’ battery levels. E-UCB1, energy aware Kullback libeler UCB (E-KLUCB), and E-TS were proposed with improved system performance. Moreover, in [38] we extended the problem solution using E-MOSS algorithm. Different from our previous works given in [4,5,38] handling mmWave NDS using noncontextual MABs, we reformulate the problem using contextual MABs while leveraging the Wi-Fi information as context in the current work. We will prove the potency of the proposed contextual-based algorithms over the noncontextual ones due to the valuable Wi-Fi contextual information. The authors of [39] proposed an adaptive TS (ATS) algorithm for beam alignment of mmWaves. ATS can precisely evaluate the best beam/rate pair without assuming any channel settings and user mobility. However, their main contribution was in beam alignment, not D2D NDS.

Contextual bandits have been applied for fundamental areas in wireless communications [40] like machine type communications (MTC) [41], cooperative communications [42], link adaptation [43], and wireless handover optimization [44]. This motivates us to leverage CMAB to solve D2D NDS critical problem, especially with the challenging difficulties of mmWaves. Although some related work contained context-aware algorithms, this paper is inspired by Wi-Fi signal’s merits, such as ease of obtainability with low latency and relative relation to mmWave signal strength.

## 3. System Model

This section presents the considered system model plus the utilized Wi-Fi and mmWave link models, including the mmWave blockage model. Moreover, the optimization problem of mmWave D2D NDS will be formulated followed by a brief discussion about the CMAB concept.

### 3.1. Multiband D2D Network Architecture

Figure 1 shows the network planning of the multiband (mmWave/Wi-Fi) D2D communication network, where multiband devices, like QUALCOMM and Intel triband devices [13,14], are uniformly located within the 4G/5G LTE-based base station (BS) (e.g., femtocell) allocated zone. Multiband D2D connections can enhance the BS coverage and its traffic offloading. The 4G/5G LTE BS will deliver the necessary signaling to supervise the mmWave D2D communication operation, including the devices remaining energies and transmission characteristics. Moreover, it handles D2D broadcasting demands, changing between cellular and D2D modes, movement supervision, and network caching. Therefore, the processing of separate D2D links, including NDS, are completed using the spread devices. In a conventional direct NDS scheme, the central device attempts careful adjacent devices exploration by recurrently doing exhaustive search BT with all the surrounding devices to attain the finest transmit/receive (TX/RX) beam pairs for reliable linking. This is performed by accounting both LOS and non-LOS (NLOS) routes originated from obstructions, see Figure 1. Subsequently, the nearby device that owns the highest data rate in Gigabit per second (Gbps) is chosen for the mmWave D2D linkage setup. Conventional NDS scheme requires a considerable BT overhead, profoundly influencing the mmWave D2D network performance.

Furthermore, most exiting NDS schemes neglected the lasting energies of the adjacent devices while carrying out NDS. That is, the selected device may not have enough energy for conducting the D2D functionality. Instead, in this paper, we will make use of the Wi-Fi information in initializing mmWave D2D NDS procedure. The solid relative relationship between Wi-Fi and mmWaves link statistics, as given in [3,14,15,16,17,18], along with the previous works of [45,46,47] that efficiently made use of Wi-Fi information to efficiently handle mmWave challenges inspired us to use Wi-Fi information as context. Thus, CMAB is best fitted to this problem besides reflecting the residual energies of the nearby devices. In our scenario, the mmWave devices are usually stationary or slow motion close to the individual’s speed. Hence, device mobility is left for future studies.

### 3.2. Wi-Fi Linkage Model

Regarding the Wi-Fi model, we will utilize the linkage model provided in [3,16,17], where the Wi-Fi received power $P_{r}^{w}$ at a reference distance *r* between two devices functioning at 5.25 GHz (Wi-Fi band) is formulated as [3]:(1)$$P_{r}^{w}\left[dBm\right]=P_{t}^{w}\left[dBm\right]-47.2-10\eta_{w}\;{log}_{10}\left(r\right)-\chi_{w},\;$$
where $P_{t}^{w}$ and $P_{r}^{w}$ are the transmitting and receiving Wi-Fi powers in dBm, respectively. Path loss exponent is $\eta_{w}$= 2.32, and $\chi_{w}∽N\left(0,\sigma_{w}\right)$ is the Wi-Fi log-normal shadowing with zero mean and 6 dB standard deviation, i.e., $\sigma_{w}\;$= 6 dB [3].

### 3.3. mmWave Linkage and Blockage Models

For the mmWave model, the mmWave received power, Prm, bearing in mind beamforming gain and blockage effects, from an adjacent device located at a distance *r* can be expressed as [3,4]:(2)Prm=PtmΛTX(ϑ)ΛRX(φ)(η(ℙLOS(r))LmLOS(r)+β(ℙNLOS(r))LmNLOS(r))
where η(ℙLOS(r)),β(ℙNLOS(r)) are Bernoulli random variables (RVs) that reflect the blockage effect with parameters ℙLOS(r), ℙNLOS(r) that indicate the distance-dependent LOS and NLOS probabilities; where ℙNLOS(r)=1−ℙLOS(r). Ptm is the mmWave TX power and ΛTX(ϑ) and ΛRX(φ) are the transmitting and receiving beamforming gains as functions of the angle of departures (AoD), i.e., ϑ, and the angle of arrival (AoA), i.e., φ. Lmv(r); where v∈{LOS,NLOS} is the distance-dependent path loss formulated in dB as [3,4]:(3)10log10(Lmv(r))=βmv+10ηmvlog10(r)+χmv, 
where βmv=82.02−10ηmvlog10(r0) is the reference path loss at the reference distance r0 = 5 m. ηmv identifies path loss exponent, and χmv∽N(0,σmv) indicates the log-normal shadowing with zero mean and standard deviation of σmv.

Regarding $\Lambda_{TX}\left(\vartheta \right)$, the 2D steerable antenna formula with Gaussian main loop shape provided in [3,4,6] is utilized, which is modeled as:(4)$$\Lambda_{TX}\left(\vartheta \right)=\Lambda_{0}e^{-4\ln\left(2\right){\left(\frac{\vartheta }{\vartheta_{-3\mathrm{dB}}}\right)}^{2}},\;\Lambda_{0}={\left(\frac{1.6162}{\sin\left(\frac{\vartheta_{-3\mathrm{dB}}}{2}\right)}\right)}^{2}$$
where ϑ, $\vartheta_{-3\mathrm{dB}}$ and $\Lambda_{0}$ represent the azimuth angle, −3 dB beamwidth, and maximum antenna gain, respectively. The same equation is applied for evaluating $\Lambda_{RX}\left(\varphi \right)$ except that RX and φ are used instead of TX and ϑ, respectively.

For mmWave blockage, we utilize the blockage scenario presented in [48], which is appropriate for both indoors and outdoors. In this scenario, mmWave obstructions are represented as cylinders that follow 2D homogenous Poisson point process (PPP) in its spatial distribution. Hence, ℙLOS(r) is expressed as [6]:(5)ℙLOS(r)=ge−ωr, 
where g=e−πΔλE[Ω2] and ω=2ΔλE[Ω], λ represents the obstacles density, Δ, Ω are the cylinder’s thinning factor and radius, respectively. E[.] is the mean operator.

### 3.4. mmWave D2D NDS Problem Modeling

The main aim of the mmWave D2D NDS process is to maximize the D2D link’s long-term average throughput/reward by considering the remaining battery levels of the distributed nearby devices. Such maximization problem is outlined as:$$\underset{1\leq i\leq N}{\max}E\left(\Psi_{i,t}\right)\;$$
s.t.
(6)$$\Xi_{i,t}>\Xi_{\mathrm{limit}}\;\mathrm{given}\;X_{i,t}\;\mathrm{for}\;\mathrm{each}\;\mathrm{device}\;i$$
where N specifies the number of the adjacent devices. $\Psi_{i,t}$ reflects the D2D linkage throughput in Gbps with adjacent device i at round t. Here, t points to the time instance of the mmWave D2D linkage request. In NDS process, the next round comes when new frames need to be sent. More precisely, the central device data is fragmented into frames and at every frame duration an NDS decision is taken to select the most appropriate nearby device to transmit its data.$\;\Xi_{i,t}$ reflects the remaining energy of the adjacent device i at instant t in joule, and $\Xi_{\mathrm{limit}}$ defines the limited energy threshold within the device for keeping its primary activities.$\;\;X_{i,t}$ is the Wi-Fi context information vector of length *d* for device i at a time t. $\Psi_{i}\left(t\right)$ formula is given as:(7)$$\Psi_{i,t}=W_{m}\Upsilon_{i,t}\left(\frac{T_{D}}{V_{t}T_{BT}+T_{D}}\right),$$
where $W_{m}$ designates the utilized mmWave bandwidth, $T_{D}$ is the required time for data transmission, $T_{BT}$ represents the BT time consumed by the central device to explore only one of its adjacent devices. $V_{t}$ reflects the number of adjacent devices performing BT with the center device at instant t. Hence, $V_{t}\;\mathrm{always}\;\mathrm{equals}\;N$ in the conventional direct NDS scheme.$\;\Upsilon_{i,t}$ represents the D2D linkage’s SE in bps/Hz related to adjacent device i at instant t, formulated as:(8)$$\Upsilon_{i,t}=\log_{2}\left(1+\frac{P_{r_{i,t}}^{m}}{N_{0}}\right),\;$$
where $P_{r_{i,t}}^{m}$ is the mmWave power received by nearby device i at instant t, and $N_{0}$ reflects the receiver’s noise power. Assume that each arm has a feature vector $X_{i,t}\;$∈ R*^d^*, which is the Wi-Fi information in our case, expressed as:(9)$$X_{i,t}={\left[x_{1_{i,t}},\;x_{2_{i,t}},\;x_{3_{i,t}}\right]}^{T}\;x_{1_{i,t}}=P_{r_{i,t}}^{w},\;x_{2_{i,t}}=E\left[P_{r_{i,t}}^{w}\right],\;x_{3_{i,t}}=var\left[P_{r_{i,t}}^{w}\right]$$
where ${\left[\right]}^{T}$ means transpose, and $P_{r_{i,t}}^{w}$ is the instantaneous received Wi-Fi power at nearby device *i* from the central device at time t. $E\left[P_{r_{i,t}}^{w}\right]$ and $var\left[P_{r_{i,t}}^{w}\right]$ are its average value and variance up to instant t. CMAB adopts the concept that the predictable reward of an arm *i* is linear with respect to its feature vector. Thus, to implement the proposed algorithm, the expected reward of arm/device *i* is proposed to be linear in its *d* dimensional context feature vector $X_{i,t}$ with unknown coefficient vector $\theta_{i}^{*}$ for all *t*, which is given as [20,21]:(10)$$E[rw_{i,t}|X_{i,t}]=X_{i,t}{}^{T}\theta_{i}^{*},$$

The CMAB game aims to estimate $\theta_{i}^{*}\;$given $X_{i,t}{}^{T}$ through successive online training.

### 3.5. CMAB Concept

To solve the optimization problem in (6), we leverage a proper type of bandits called CMAB. where, the player accumulates her rewards from taking actions (selecting arms) over a sequence of trials. During each round, the player takes action upon both contexts (feature vector) for the current round and the previously collected rewards obtained in the previous trials. The player notices the reward only for the chosen arm. CMAB exists in several vital applications like online recommendations, mobile health applications, and clinical trials [43]. The feature utilization to encode context is acquired from supervised ML, while exploration is vital for improving the learning performance like RL technique. Hence, CMABs is the usual halfway argument between supervised learning and RL [49]. Usually, the CMAB problem is solved via proposing a linear relationship between the produced reward and its related contexts as given in (10) and addressed by LinUCB [20] and CTS [21] algorithms.

The standard CMAB problem can be formulated as follows. Let A={1,…,N} be the set of *N* existing independent devices/arms. Let X⊆ℝd be a set of *d*-dimensional context vectors that depict players/devices and their surroundings, i.e., each member is a binary vector encoding features such as arm locations, decisions, pursuits, etc. For each round *t* ∈ [1, *T*] and each arm i∈A, the context vector, Xi,t∈X, is given to the algorithm from the environment to select an arm. Assume that rwt=(rwi,t,…rwN,t) is the reward vector at trial *t*, where rwi,t is the collected reward via selecting arm/device *i* at round *t* that follows some unknown Gaussian distribution in our case. *θ_i_* is an unknown coefficient vector (to be learned) related to arm *i* at round *t*. An assumption is made that the expected rewards of an arm/device *i* at trial *t* is linearly related to the *d*-dimensional context vector Xi,t as given in (10). The general CMAB protocol is summarized in Figure 2.

## 4. Proposed EA-CMAB Algorithms

Herein, we will discuss two proposed EA- CMAB algorithms that handle mmWave D2D NDS proficiently. In our setting, single-player CMAB is concerned, and multiplayer CMAB scenario will be left for future investigations. First, we will explain the device’s battery update equation followed by the proposed EA-LinUCB and EA-CTS algorithms. At every round *t*, the proposed CMAB algorithm will select a nearby device, $i_{CMAB}^{*}$, where its updated residual energy, $\Xi_{i_{CMAB}^{*},t}$, is given by: -
(11)$$\Xi_{i_{CMAB}^{*},t}=\Xi_{i_{CMAB}^{*},\;t-1}-\frac{P_{t}^{m}L_{D}}{W_{m}\;\Upsilon_{i_{CMAB}^{*},t}}$$
where $\Xi_{i_{CMAB}^{*},\;t-1}$ is its remaining energy at instant t−1. The expression $P_{t}^{m}L_{D}/W_{m}\;\Upsilon_{i_{CMAB}^{*},t}$ reflects the consumed energy to fetch the necessary $L_{D}$ data bits with $W_{m}\;\Upsilon_{i_{CMAB}^{*},t}$ bps data rate by the selected nearby device $i_{CMAB}^{*}$. An assumption is made that all devices have equal transmit powers to fetch data.

The modified algorithms take into account the remaining energy levels of the nearby devices during their arm selection. This is done by appending the energy term,$\;\rho \frac{r_{i,t}}{\Xi_{i,t}\;}$, in the main exploration part of each algorithm to reflect the real scenario where some devices may run out of their energy and be excluded from the game. The added term compromises between the obtained throughput and consumed energy of each selected device. Hence, the EA-CMAB algorithms will choose the highest energy and largest throughput device among others. Therefore, the algorithms will not be stuck to the lowest energy or the highest throughput device.

### 4.1. Proposed EA-LinUCB Algorithm

LinUCB [20] extends the Auer’s UCB algorithm in [10,22] to the contextual concept. Its main clue is to figure out each arm’s probable reward by finding a linear relationship between the previous rewards of the arm and its current context vector as given in (10). LinUCB interprets the features vector of the existing round into a linear combination of features vectors seen on former rounds and utilizes the calculated coefficients and rewards on earlier rounds to calculate the anticipated reward on the present round. Let $G_{i}$ be an m×d matrix at trial *t*, whose rows represent m contexts noticed previously for arm/device *i*. Applying ridge regression to the training data ($G_{i},b_{i}$) gives an estimate of the coefficients:(12)$${\hat{\theta }}_{i}={\left(G_{i}^{T}G_{i}+I_{d}\right)}^{-1}\;b_{i}$$
where $b_{i}=G_{i}^{T}c_{i}$, where $c_{i}\;$is the m-dimensional vector whose components are past observed rewards of arm *i.* When the $c_{i}$ components are independently conditioned on corresponding rows in $G_{i}$, it can be shown that [20]
(13)$$\left|X_{i,t}{}^{T}{\hat{\theta }}_{i,t}-E[\Upsilon_{i,t}|X_{i,t}]\right|\;\leq \alpha_{LinUCB}\sqrt{X_{i,t}{}^{T}\;B_{i}{}^{-1}X_{i,t}}$$
where$\;B_{i}=G_{i}^{T}G_{i}+I_{d}\;\mathrm{and}\;\alpha_{LinUCB}=1+\sqrt{\ln\left(2/\delta_{\mathrm{LinUCB}}\right)}\;\mathrm{for}\;\delta_{\mathrm{LinUCB}}>0.{\;\Upsilon }_{i,t}\;\mathrm{is}\;$the SE/reward of drawing arm/device *i* at round t calculated from (8). The above inequality provides a reasonable strong UCB for the expected reward of device *i*. Similar to UCB arm selection strategy, at each trial *t*, the best arm $i_{t}^{*}$ is selected as follows:$$i_{t}^{*}=\arg\;\underset{i\in A}{\max}\;\left(\;j_{i,t}\right),$$
where
(14)ji,t=Xi,tTθ^i,t+αLinUCBXi,tT Bi−1Xi,t−ρri,tΞi,t  
where ri,t is the distance of device *i* from central device at instant t. The new term ρri,tΞi,t  is added to the standard LinUCB equation to mirror the remaining energies of the spread devices upon their locations from the central device. That is, for a constant data length LD in (11), higher remaining energy is required by a faraway device to establish the D2D linkage owing to the reduction of its attainable data rate and vice versa. Algorithm 1 provides the detailed explanations of the proposed EA-LinUCB algorithm. The inputs are the threshold energy limit, Ξlimit, and the energy of the adjacent devices at t=1 plus the parameter αLinUCB. The arms having higher remaining energies than Ξlimit will be involved in the game. After applying the EA-LinUCB, the parameters are updated for the next round when new data frames need to be sent by the central device, as given in Algorithm 1.
**Algorithm 1**: EA-LinUCB NDS**Input:**Ξlimit and Ξi.1 for ∀ i∈A,  αLinUCB∈ℝ+ **For** t =1, 2,…, T     Notice features of ∀ i∈A: Xi,t∈X⊆ℝd    **For** ∀ i∈A do    **While**
Ξi.t> Ξlimit     **If** arm *i* is new then      Bi=Id (identity matrix)       bi=0d×1 (zero vector)     **End If**           θ^i=Bi−1bi          ji,t=Xi,tTθ^i,t+αLinUCBXi,tT Bi−1Xi,t−ρri,tΞi,t      **End While**    **End For**  Choose arm it∗=argmaxi
(ji,t) and observe its reward Υi∗,t from (8)     1.
Bit∗=Bit∗+Xit∗,t Xit∗,tT       2. bit∗=bit∗+ Υit∗Xit∗,t       3. Ξi∗,t+1=Ξi∗,t−(PtmLDWm Υit∗)   **End For**

### 4.2. Proposed EA-CTS Algorithm

TS [23] fundamental policy applies Bayesian strategy because the rewards are supposed to be pulled upon a known probabilistic model. A simple former distribution is suggested for the rewards of each arm based on parameter initialization. Then, within the learning process, the TS strategy updates the rewards’ posterior distribution using the collected data to draw the optimal probable arm. Precisely, at every round t, random samples are drawn from the rewards’ posterior distributions, then the arm having the highest sample value is chosen. Afterward, the chosen arm’s posterior distribution is updated for the upcoming round of arm choice. For CTS, we assume a slightly simpler model on the CMAB protocol given in Figure 2, where the main difference is that we assume that there exists θ s.t. $\theta_{i}=\theta$ for all arms i∈A. The global construction of CTS for the CMAB problem includes the subsequent fundamentals [21]:A set θ of parameters $\tilde{\theta }$.A former distribution *P*($\tilde{\theta }$) which is Gaussian in our case.Former observations, *D,* containing (context *X*, reward Υ) for the previous time steps.$P(\Upsilon |X,\tilde{\theta }),\;$the probability of reward Υ given a context *X* and a parameter $\tilde{\theta }$.Posterior distribution *P*($\tilde{\theta }$|*D*) ∝ *P*(*D*|$\tilde{\theta }$)*P*($\tilde{\theta }$).

At each round *t*, CTS pulls an arm upon its posterior probability. This simply can be done by taking a sample from each arm via the posterior distributions and selecting the arm with the best sample. Because the reward distribution is Gaussian due to the Gaussian noise, we utilize the Gaussian likelihood function and Gaussian prior for our EA-CTS. Expressly, assume that the likelihood of reward $\Upsilon_{i,t}$ at time *t*, given context $X_{i,t}\;$are provided from the normal distribution $\left(X_{i,t}^{T}\tilde{\theta },\;\partial_{CTS}^{2}\right)$, where $\partial_{CTS}=R\sqrt{\frac{24}{\epsilon }\;d\;\ln\left(\frac{1}{\delta_{CTS}}\right)}$ with ϵ∈ (0, 1). Let
(15)$$B_{t}=I_{d\;}+\overset{t-1}{\underset{h=1}{\sum }}X_{i_{h},h}X_{i_{h},h}^{T}\;$$
(16)$${\overset{´}{\theta }}_{t}=B_{t}^{-1}\overset{t-1}{\underset{h=1}{\sum }}X_{i_{h},h}\Upsilon_{\;i_{h},h}$$

Then, if the prior distribution of θ at time *t* is known as $N\left(\overset{´}{\theta_{t}},\;\partial_{CTS}^{2}B_{t}^{-1}\right)$, then the posterior distribution at time *t* + 1 is given as $N\left(\;{\overset{´}{\theta }}_{t+1},\partial_{CTS}^{2}\;B_{t+1}^{-1}\right)\;$ [21]. Our modified algorithm produces a sample ${\tilde{\theta }}_{t}\;$from $N\left(\overset{´}{\theta_{t}},\;\partial_{CTS}^{2}B_{t}^{-1}\right)$ distribution and selects the arm maximizing $X_{t,i}^{T}{\tilde{\theta }}_{t}\;$−$\;\rho \frac{r_{i,t}}{\Xi_{i,t}\;}$. Herein, we utilize Gaussian-based EA-CTS because of Gaussian distribution of the reward as shown in [4]. Algorithm 2 summarizes the EA-CTS main steps, where the first step is to select the devices with high remaining energies inside the selection range. Then the algorithm produces a *d*-dimensional sample ${\tilde{\theta }}_{t}$, from a multivariate Gaussian distribution, and attempts to solve the maximization problem $\arg\underset{i\in A}{\max}\left(X_{t,i}^{T}{\tilde{\theta }}_{t}-\rho \frac{r_{i,t}}{\Xi_{i,t}\;}\right)$. As given in EA-LinUCB, the newly added term, $\rho \frac{r_{i,t}}{\Xi_{i,t}\;},$ reflects the remaining energies of the surrounding devices. The parameters $B,f,\tilde{\theta },{\;\Xi }_{i^{*}}$ are updated for next round selection to send new data frames as given in Algorithm 2.
**Algorithm 2**: EA-CTS NDS**Let**$B=I_{d\;},\tilde{\theta }=0_{d},f=0_{d}$ **For**t=1,2,....T   **While**$\Xi_{i.t}>{\;\Xi }_{limit}$    Sample ${\tilde{\theta }}_{t}$, from normal distributions $N\left(\overset{´}{\theta_{t}},\partial_{CTS}^{2}B_{t}^{-1}\right)$    **Play arm**$i_{i,t}^{*}\in \;\arg\underset{i\in A}{\max}\;\left(X_{t,i}^{T}{\tilde{\theta }}_{t}-\rho \frac{r_{i,t}}{\Xi_{i,t}\;}\right)\;$ and notice     the reward $\Upsilon_{i_{t}^{*}}$**,** i.e., SE obtained from (8)   **Update**     1. $B=B+X_{t,i_{t}}X_{t,i_{t}}^{T}$     2. $f=f+X_{t,i_{t}}\Upsilon_{i_{t}^{*}}$     3. $\tilde{\theta }=B^{-1}f$     4. $\Xi_{i^{*},t+1}=\Xi_{i^{*},t}-\left(\frac{P_{t}^{m}L_{D}}{W_{m}{\;\Upsilon }_{i_{t}^{*}}}\right)$    **End While** **END For**

## 5. Numerical Results

This section presents the conducted numerical simulations that confirm the superior performance of the proposed EA-CMAB-based algorithms using 10,000 rounds of Monte Carlo (MC) simulations throughout MATLAB environment. Every MC round includes randomized device locations, randomized channel properties (mmWave and Wi-Fi related shadowing terms), randomized mmWave blocking patterns coming from the tested blocking probability, and randomized battery initialization of each distributed nearby device. To approve that, the proposed algorithms are compared with mostly related noncontextual solutions [4,5], besides the famous traditional selection techniques, named conventional and random selection schemes. The conventional NDS scheme searches all devices before deciding the best one, which consumes a considerable time and achieves a significant BT overhead. However, in random NDS, the adjacent device is picked randomly from the surrounding devices at every round *t* to establish the mmWave D2D link. The total average throughput is evaluated by averaging (7) over the game’s time horizon *T*. Hence, $V_{t}=N$ for conventional selection scheme, while for CMAB proposed algorithms and random scheme $V_{t}=1$. The EE is formulated as:(17)$$EE=\frac{1}{N}\overset{N}{\underset{i=1}{\sum }}\Psi_{i}\left(T\right)/\left(\Xi_{i,1}-\Xi_{i,T}\right)$$
where $\Xi_{i,1}$is the device *i’s* starting energy and $\Xi_{i,T}$ reflects its final energy when the game is terminated. Table 1 summarizes the related simulation parameters, where around 20 to 100 devices are uniformly diffused in a region of 125 × 125 m^2^. Moreover, ideal beam alignment is considered within the D2D devices, i.e., $\Lambda_{TX}\left(\vartheta \right)=\Lambda_{RX}\left(\varphi \right)=\Lambda_{0}$.

### 5.1. Without Battery Consideration

Herein, we will figure out the merits of CMAB algorithms over noncontextual ones in mmWave NDS. Figure 3 shows the average throughput versus the number of distributed devices at no blocking (λ = 0) for UCB, TS, LinUCB, and CTS algorithms. The two CMAB algorithms’ performance is close to each other due to the utilized stationary scenario shown in [50,51] and the Wi-Fi context vector, not the mmWave-based one. The noncontextual MAB algorithms (UCB and TS) show improved performances over conventional and random selection methods. The CMAB algorithms (LinUCB and CTS) have a superior performance that is close to the optimum, where the optimal NDS performance comes via selecting the best device having the maximum SE from the first time, i.e., $V_{t}=1$. In conventional direct NDS scheme, the throughput is reversely related to the number of surrounding devices because the exhaustive BT produces considerable overhead. The other compared schemes (optimal, LinUCB, CTS, TS, UCB, and Random) have small BT overhead due to performing BT with a single device every round. However, the random scheme experiences the worst performance due to the adjacent device randomization selection policy. It is interesting to notice that the throughputs of the LinUCB and CTS schemes are improved relatively with increasing the number of devices because of the valuable context vector that maximizes long-term throughput with small BT overhead. CMAB performance is higher than TS and UCB, which indicates the effectiveness of the contextual information. At 40 (80) devices, about 96.3% (97.4%), 94.7% (91%), 82.6% (67.24%), 80.5% (43.1%), and 59.3% (48.3%) of the optimal performance are obtained using LinUCB/CTS, TS, UCB, conventional and the random schemes, respectively.

Figure 4 presents the average throughput performances of the examined methods using 60 devices versus various blocking densities, i.e., changing values of λ. As blocking is enlarged, the average throughput of all methods decreases because of the increased NLOS probability (blockage) that decreases the received power and hence, the attainable data rate. The random scheme also yields the most defective throughput performance due to the randomized device selection policy that may experience abrupt blocking. However, the CMAB-based NDS displays near optimal performance. At λ of 0 (0.15) about 96.9% (95.6%), 92.8% (90.8%), 81.6% (80.8%), 56.2% (54.2%), and 51.3% (20%) of the optimal performance are obtained using LinUCB/CTS, TS, UCB, conventional and the random schemes, respectively.

Figure 5 shows the convergence rate of the LinUCB, CTS, UCB, and CTS algorithms against optimal and random performances. It is worth noting that the convergence of TS is faster than UCB due to the Bayesian policy of TS. At t=100, both LinUCB and CTS converge to around 98% of the optimal throughput, while noncontextual bandits own slower convergence, where TS converges to 91% while UCB converges to 73%.

### 5.2. With Battery Consideration

Figure 6 shows the average throughput performances against the number of distributed devices at no blocking (λ = 0). The proposed EA-CMAB algorithms (i.e., EA-LinUCB and EA-CTS) show better performance than not only similar noncontextual ones (i.e., EA-UCB and EA-TS [4]) but also conventional and random selection schemes too. Both EA-CMAB schemes have close performance due to the close performance of both LinUCB and CTS as previously explained, plus the newly added energy term [50,51]. The average throughput performance of EA-LinUCB and EA-CTS schemes are increased proportionally with the number of devices due to the effective Wi-Fi context vector that increases the long-term reward and reduces the BT cost. At 20 (100) devices, both EA-CMAB algorithms have 1.3 (5.5) and 2.8 (5) throughput improvement against conventional and random selections, correspondingly. The two modified EA-CMAB algorithms display similar performance, showing small throughput fluctuations affected by the lately appended remaining energy expression, i.e., $\frac{r_{i}}{\Xi_{i,t}}$. This expression affects the typical CMAB algorithms’ estimation by prioritizing closer devices, reaching more excellent realizable data rates with lower consumed energies. Moreover, EA-CMAB shows higher performance than noncontextual EA-MAB algorithms.

Figure 7 shows the throughput evaluation of the related methods versus different blocking values via 60 devices. All schemes’ throughput is inversely related to the blocking density values. Moreover, the random selection also displays the most deficient performance. However, the proposed EA-CMAB-based NDS exhibits near-optimal performance because of the optimized chosen device with the help of Wi-Fi information. At blocking densities of 0 (0.15), the EA-CMAB (EA-LinUCB and EA-CTS)-based NDS has 0.5 (0.5), 4 (3.8) and 4.2 (8) throughput performance increments over EA-MAB (EA-UCB and EA-TS), conventional and random schemes, respectively. Moreover, EA-CMAB outperforms EA-MAB algorithms.

Figure 8 displays EE performances in Gbps/mJ against the number of distributed devices at no blocking (λ = 0). The EEs of all compared schemes are increased relatively with increasing the number of nearby devices because of the large number of devices having higher SE values available for setting up the mmWave D2D linkage. This intensely reduces the spent energy of the chosen device in accordance. Furthermore, random selection reveals the worst performance, while the two EA-CMAB-based NDS algorithms display better performance than EA-MAB algorithms. Due to the additional energy-constraint to the formulated CMAB problem, the EA-CMAB-based NDS maximizes the long-term throughput while conserving the adjacent devices’ remaining energies when constructing the D2D links through making use of Wi-Fi contexts. This improves EE performances over both noncontextual EA-MAB and the conventional and random NDS. At 20 (100) devices, the EA-CMAB-based NDS has 0.1 (0.5), 1.3 (2.5) and 2 (3.5) increase in EE over EA-MAB, conventional and random schemes, respectively.

Figure 9 demonstrates the EE evaluations versus different blocking λ values using 60 devices. Generally, as λ is increased, the EE of all algorithms is decreased. This is due to the significant blockage effect, which reduces the available data rate extending data transmission time resulting in more considerable energy dissipation as given in (9). Still, the proposed EA-CMAB algorithms show the best EE performances within whole λ values because of the context vector’s influence and the energy constraint. However, the random scheme demonstrates the most defective EE values at different values of λ. At blocking densities of 0 (0.15), the EA-CMAB-based NDS has 1.5 (1.7) and 2.9 (29) increments in EE over conventional and random schemes, accordingly.

Figure 10 illustrates the convergence comparisons of EA-LinUCB and EA-CTS algorithms versus EA-MAB (EA-UCB and EA-TS), random, and optimal schemes. For the sake of comparison, the optimal scheme is by considering device’s infinite energy. EA-CMAB converges faster than EA-MAB algorithms, resulting in faster learning process. Nearly at 100 rounds, the two proposed EA-CMAB algorithms converge to 99% of the optimum average throughput, while EA-MAB convergence equals 96%. EA-CMAB algorithms have slight faster convergence than EA-MAB schemes, which ensures its appropriate selection for the problem solution.

For complexity analysis, the time consumed by the compared schemes comes from algorithm execution time and nearby device probing time. The execution time of the proposed CMAB algorithms is of order $O\left(d^{2}N\right)$ [20,21], which greatly depends on the number of the probed devices and size of the context vector d. Regards of *d,* it is fixed to three as previously explained, and *N* is a small value because we only consider the scenario of a small cell with a few numbers of surrounding users. Moreover, according to our proposed algorithms policy, *N* decreases with the trials increment because of the battery condition. Hence, our algorithm’s processing time can be considered as constant, especially at small nearby devices case. In Table 2, we measured the MATLAB R2020 b execution time of the proposed algorithms against the number of devices compared to the conventional scheme. The specifications of the used machine are i7-8565U CPU @ 1.80 GHz 1.99 GHz and 8 GB RAM. From Table 2, the execution time of the proposed algorithms are in the range of milliseconds which fit the 5G/6G requirements of millisecond latency. Moreover, typically, MATLAB software consumes large execution time because of its complier. Hence, we expect much lower execution time compared to these values when implemented in real hardware platforms. The second source is the BT time of one device probing which is about 0.28 msec [1]. This ensures the near optimal performance of the proposed CMAB/EA-CMAB schemes as given in Figure 3, Figure 4, Figure 5, Figure 6, Figure 7, Figure 8, Figure 9 and Figure 10.

## 6. Conclusions

This paper discussed resolving the NDS problem in mmWave D2D communications using ML-based CMABs. It advanced a CMAB-based online learning technique that effectively solves the NDS problem for future talented applications. This is done by making use of Wi-Fi information of the nearby multiband standardized devices as context information. Hence, LinUCB and CTS schemes were leveraged for NDS solution and their performance was investigated against UCB and TS algorithms. Afterward, we proposed EA-LinUCB and EA-CTS to accelerate the discovery process and take full advantage of the long-term average throughput while bearing in mind the remaining energies of the adjacent devices. The suggested algorithms confirmed their superior performances, which are higher than noncontextual MAB algorithms plus traditional mmWave D2D NDS approaches. EA-CMABs achieved larger EE than other schemes with faster convergence rates. Future research directions will be directed towards practical experimental implementations and multiplayer scenarios using CMABs in centralized and decentralized settings. Moreover, implementing deep CMABs looks a promising approach.

## Figures and Tables

**Figure 1 sensors-21-02835-f001:**
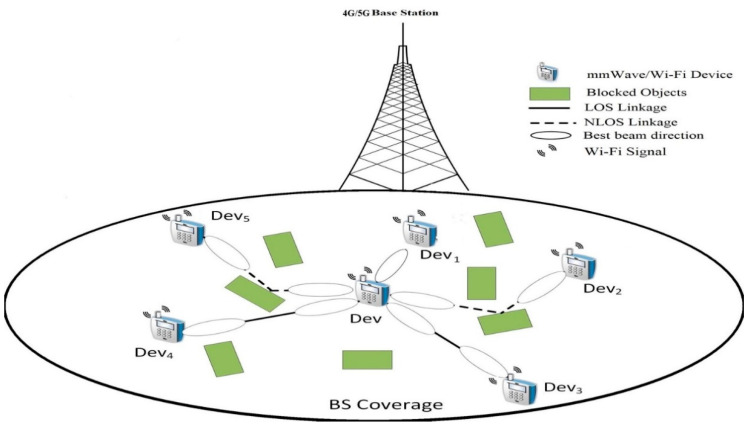
Multi-band D2D network architecture.

**Figure 2 sensors-21-02835-f002:**
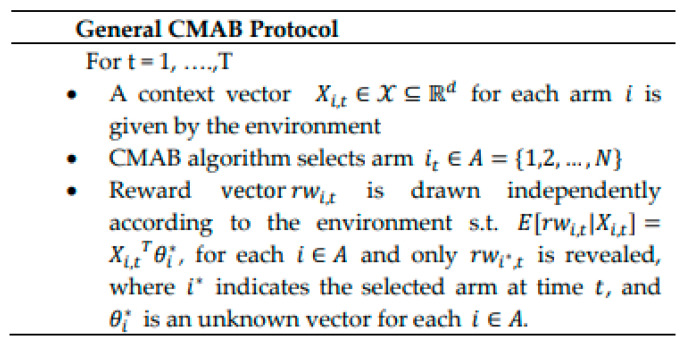
General CMAB protocol.

**Figure 3 sensors-21-02835-f003:**
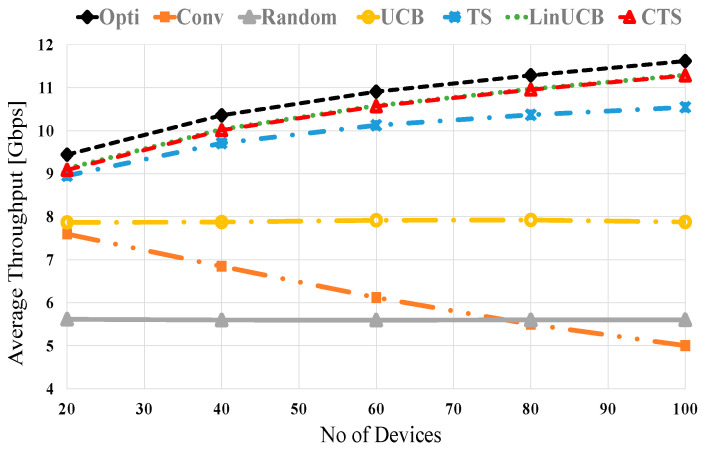
Average throughput versus the number of devices at no blocking for UCB, TS, LinUCB, and CTS algorithms.

**Figure 4 sensors-21-02835-f004:**
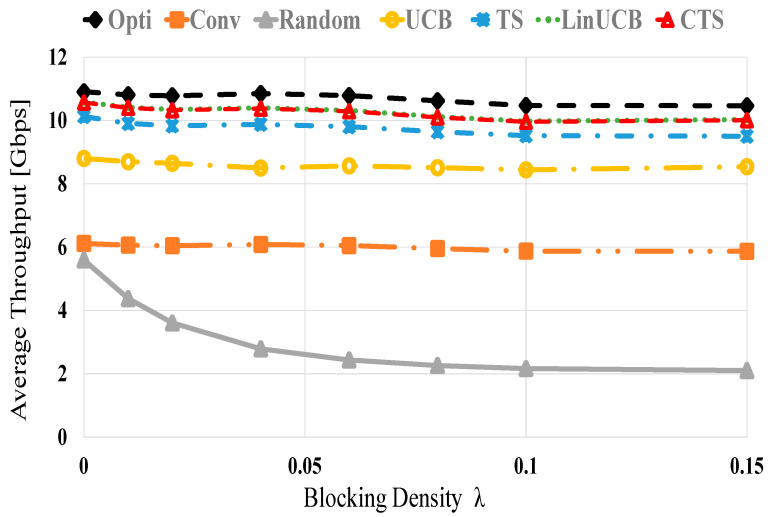
Average throughput versus blocking density λ for UCB, TS, LinUCB, and CTS algorithms using 60 devices.

**Figure 5 sensors-21-02835-f005:**
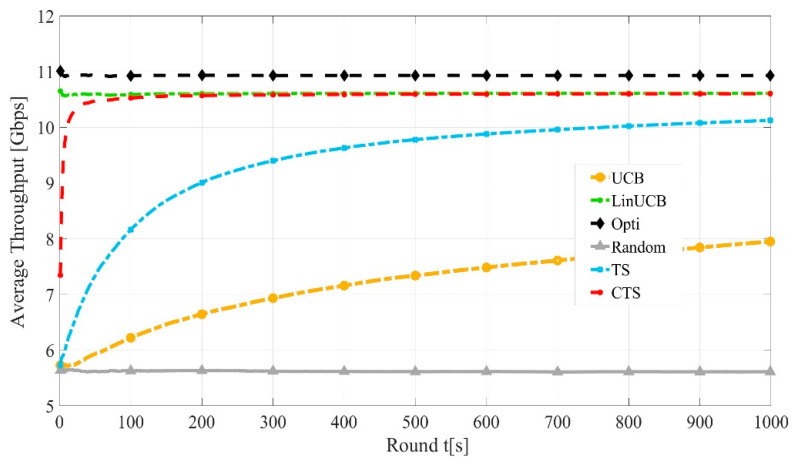
Convergence rates of LinUCB, CTS, TS, and CTS algorithms.

**Figure 6 sensors-21-02835-f006:**
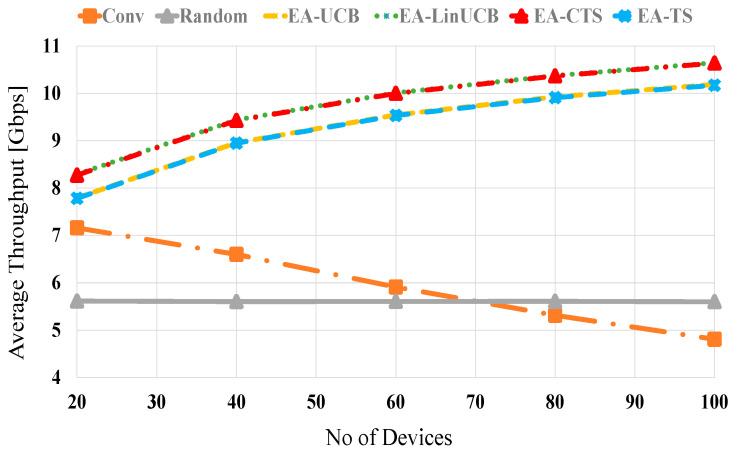
Average throughput versus the number of devices at no blocking for EA-UCB, EA-TS, EA-LinUCB, and EA-CTS algorithms.

**Figure 7 sensors-21-02835-f007:**
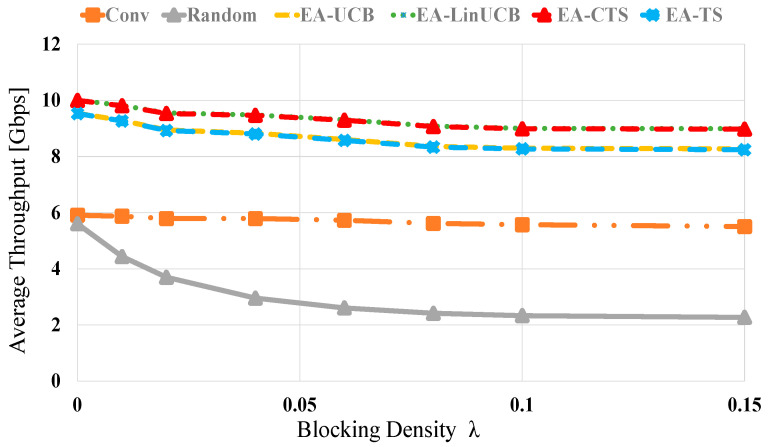
Average throughput versus blocking density λ for EA-UCB, EA-TS, EA-LinUCB, and EA-CTS using 60 devices.

**Figure 8 sensors-21-02835-f008:**
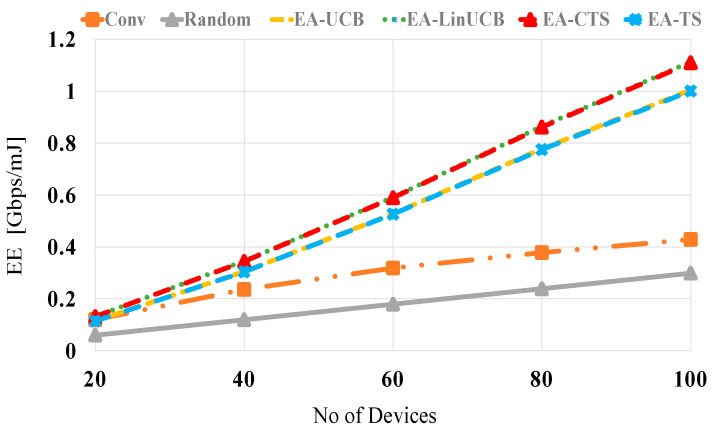
*EE* versus the number of devices for EA-UCB, EA-TS, EA-LinUCB, and EA-CTS at no blocking.

**Figure 9 sensors-21-02835-f009:**
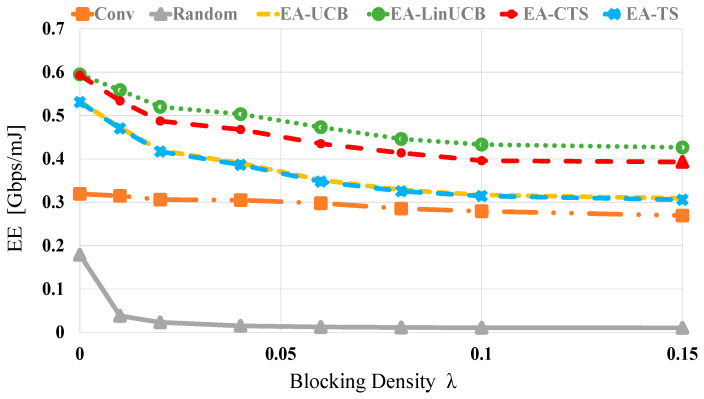
*EE* versus blocking density for EA-UCB, EA-TS, EA-LinUCB, and EA-CTS using 60 devices.

**Figure 10 sensors-21-02835-f010:**
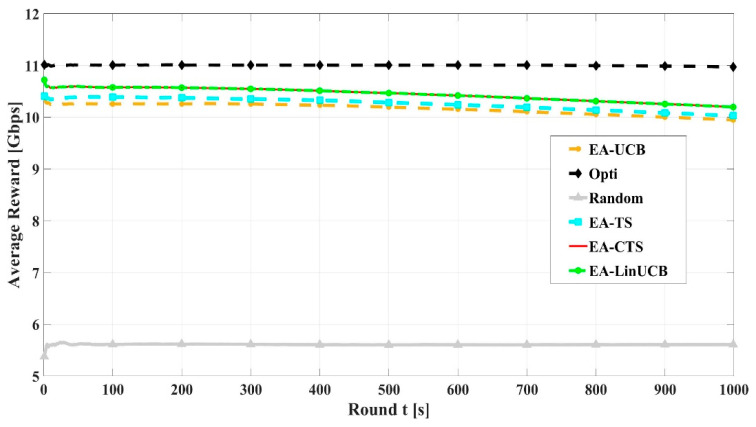
Convergence rate of energy aware algorithms using 60 devices.

**Table 1 sensors-21-02835-t001:** Simulation Parameters.

Parameter	Value
$P_{t}^{w}$ and $P_{t}^{m}$	20 and 10 dBm
$W_{m},\;T_{BT},$ and $L_{D}$	2.16 GHZ [1], 0.28 msec [1], and 1 Gbit.
$\vartheta_{-3\mathrm{dB}}$, T	20°, 1000
${\;\alpha }_{\mathrm{LOS}},{\;\mathrm{and}\;\alpha }_{\mathrm{NLOS}}$	2.22 [3] and 3.88 [3]
$\sigma_{m}^{\mathrm{LOS}}\;\mathrm{and}\;\sigma_{m}^{\mathrm{NLOS}}$	10.3 [3], and 14.6 [3]
Δ and Ω	1 and uniform [0.3–0.6] m [6]
$\Xi_{i,1}{\;\mathrm{and}\;\Xi }_{\mathrm{limit}}$	Uniform random in the range of [0.1…1] J and 0.1 J
$N_{0}$, ρ	−174 + 10log10(W) + 10, 1
$\;\alpha_{LinUCB}$, R, ϵ,$\;\delta_{CTS}$	0.4, 10^−7^, $\frac{1}{\mathrm{Lin}\;T}$, 10^−8^

**Table 2 sensors-21-02835-t002:** Execution times of the compared algorithms.

	Algorithm	EA-UCB	EA-TS	EA-LinUCB	EA-CTS	Conventional
No of Devices						
20		0.1 msec	0.2 msec	0.3 msec	0.31 msec	5.6 msec
60		0.1 msec	0.5 msec	0.6 msec	0.66 msec	16.8 msec
80		0.2 msec	0.6 msec	0.8 msec	0.9 msec	22.4 msec
100		0.2 msec	0.7 msec	0.9 msec	1 msec	28 msec

## Data Availability

Not applicable.

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
