# Peer review of "Wi-Fi Assisted Contextual Multi-Armed Bandit for Neighbor Discovery and Selection in Millimeter Wave Device to Device Communications"

_sensors, 2021, doi:10.3390/s21082835_

Round 1
Reviewer 1 Report
The authors contribute to the timely problem of neighbor discovery and selection in a B5G/6G environment. They propose to use contextual multiarmed bandit technique to select neighboring multi-band devices for mmWave D2D transmission, assuming that a WiFi signal is also available. Building on earlier reported experimental data regarding the correlation between WiFi and mmWave transmission characteristics, and the existing analytical framework, the linear upper confidence bound and Thomson sampling algorithms are applied. As part of the well-established routine, a linear transformation of the context information into current rewards is assumed. The algorithms are complemented with a residual energy constraint to maximize the energy leftover in the selected device.
The paper gives all the necessary the algorithm details and contains a thorough simulation study that confirms the superiority of the proposed algorithms to previously known. The reader notes that there is little novelty on the methodology side, but the existing algorithms have been innovatively augmented and correctly leveraged to achieve the desired energy awareness.
On the presentation side, there is a lack of more critical discussion of the assumptions made, e.g., on the availability of the WiFi signal, or access to nearby devices' energy data and transmission characteristics. I would also consider a reduction of the symbols and quantities to those that are used for the algorithm outlining and simulation, compare the "Nomenclature" list and Table 1. The body text abounds in grammatically incorrect, bizarre, or unclear statements that need to be improved. Examples: "fundamental problems amongst the neighbor discovery and selection", "A major ML branch [is] called reinforcement learning", "techniques are [more] promising solutions for mmWave communication systems than...", "game is completed once the whole devices reach...", "This paper main contributions", "Great hopes are forwarded to...", "we approved that our proposed method outperforms...", "At [to] the best 178 of our knowledge,...", "Figure 1 Figure 1" (repetition), "Though, separate D2D links establishment and ending...", "gaussian" (capitalize), "with bearing in mind...", "defines the limited energy level" (limit?), "Hence, regards the conventional direct NDS scheme,...", "which is the Wi-Fi information in our case, which can be expressed", "assumption is made that equal TX power for all devices for the sake of...", "Like the LinUCB [15] extends the Auer’s UCB algorithm...", "inequality provides a reason for the payoff of device", " A former distribution" (prior?), "Past remarks D containing (context X, reward Y)", "CTS pulls an arm at each round t along with its posterior probability..." (continuation of this paragraph is also unclear), " The policy of This section presents", "numerical analysis steered to approve the superior throughput"; there are many other examples to be corrected.
Minor remarks: in Algorithm 1, "Choose arm..." should it not be "observe Y_{i^*_t,t}"?; in Algorithm 2, argmax is used, which is ordinarily taken to be a set and not a value, if the authors understand it otherwise, an explanation should appear.
Author Response
We would like to sincerely thanks the respected reviewer for the time and efforts he/she spent in reviewing our paper. Also, we highly appreciate the valuable comments given by the respected reviewer, which we do believe it enhances the clarity and readability of the revised manuscript. In the following, we have tried our best to address all the comments given by the respected reviewer.
Kindly see the attached reply letter

Reviewer 2 Report
The paper with Manuscript ID: sensors-1137631 and title Wi-Fi Assisted Contextual Multi-Armed Bandit for Neighbor Discovery and Selection in Millimeter Wave Device to Device Communications. In this paper, the mmWave NDS problem is addressed using contextual multi-armed bandit (CMAB) algorithms. This is done by leveraging the context information of Wi-Fi signal characteristics.
Review
The paper address the relevant problem of neighbor discovery and selection (NDS) in mmWave D2D networks based on context multi-armed bandit (CMAB). The results presented are interesting versus NDS in mmWave D2D non-contextual MAB.
My main concerns.
- This paper is based on previous work published by the authors [4, 32, 33]. Not all previous papers related published by the authors are cited. The following paper must be cited and explain if the present paper is an extension of it.
@Article{electronics10020169,
AUTHOR = {Hashima, Sherief and ElHalawany, Basem M. and Hatano, Kohei and Wu, Kaishun and Mohamed, Ehab Mahmoud},
TITLE = {Leveraging Machine-Learning for D2D Communications in 5G/Beyond 5G Networks},
JOURNAL = {Electronics},
VOLUME = {10},
YEAR = {2021},
NUMBER = {2},
ARTICLE-NUMBER = {169},
URL = {https://www.mdpi.com/2079-9292/10/2/169},
ISSN = {2079-9292},
ABSTRACT = {Device-to-device (D2D) communication is a promising paradigm for the fifth generation (5G) and beyond 5G (B5G) networks. Although D2D communication provides several benefits, including limited interference, energy efficiency, reduced delay, and network overhead, it faces a lot of technical challenges such as network architecture, and neighbor discovery, etc. The complexity of configuring D2D links and managing their interference, especially when using millimeter-wave (mmWave), inspire researchers to leverage different machine-learning (ML) techniques to address these problems towards boosting the performance of D2D networks. In this paper, a comprehensive survey about recent research activities on D2D networks will be explored with putting more emphasis on utilizing mmWave and ML methods. After exploring existing D2D research directions accompanied with their existing conventional solutions, we will show how different ML techniques can be applied to enhance the D2D networks performance over using conventional ways. Then, still open research directions in ML applications on D2D networks will be investigated including their essential needs. A case study of applying multi-armed bandit (MAB) as an efficient online ML tool to enhance the performance of neighbor discovery and selection (NDS) in mmWave D2D networks will be presented. This case study will put emphasis on the high potency of using ML solutions over using the conventional non-ML based methods for highly improving the average throughput performance of mmWave NDS.},
DOI = {10.3390/electronics10020169}
}
- On page five the authors remark the following “The current work is the first that proposes ML based context-aware bandit algorithm for mmWave D2D”. First, the authors never said in the abstract nor in the introduction that they use Maching Learning in their proposal solutions (EA-CMAB, EA-LinUCB and EA-CTS). Secondly, in the abstract of the above-mentioned paper, the authors presented a case study of applying multi-armed bandit (MAB) as an efficient online ML tool to enhance the performance of neighbor discovery and selection (NDS) in mmWave D2D networks will be presented. My question is if the present paper is a direct extension of such case study? I think the authors took some parts from the previous paper and for this reason, there are some inconsistencies in the current paper.
- I think section 4 (CMBA Concept) must be part of the background and not a separate section.
- Section 5 (Proposed EA-CMAB Algorithms) must clearly specify the original contribution for each algorithm.
- In Section 6 the proposed EA-CMAB algorithms their performances are examined against two standard schemes named conventional direct NDS and random selection. Moreover, the proposed schemes are compared with the non-contextual EA-MAB (EA-UCB and EA-TS) algorithms presented in [4]. I think the comparison is not fair. First, since there are algorithms with better performance than direct NDS and random selection, and secondly, EA-UCB and EA-TS are algorithms published by the same authors.
For all these reasons my recommendation is to reconsider after major revision.
Author Response
We would like to sincerely thanks the respected reviewer for the time and efforts he/she spent in reviewing our paper. Also, we highly appreciate the valuable comments given by the respected reviewer, which we do believe it enhances the clarity and readability of the revised manuscript. In the following, we have tried our best to address all the comments given by the respected reviewer
kindly see the attached reply letter

Reviewer 3 Report
The paper proposes new algorithms for neighbor discovery and selection for a device to device communications in millimeter waves. The proposed approach exploits RF techniques to select the optimum neighbor.
The main weaknesses:
- The proposed approach is quite far from real technology. So, please provide details and on used technologies. I'm a bit confused because it is mentioned 5G, WiGi, WIFI & 5GHz, LTE. So, please make it clear what and how is actually used. Fig. 1 seems the best place to explain.
- The main assumption was to maximize the throughput between neighboring devices. It is realistic that I want to communicate with any device? Please elaborate more about the assumptions. What is for this D2D communications
- The effectiveness is evaluated based on simulations that were performed on a very simplified model. I suggest enhancing evaluation by some lab experiments that will show how the proposed algorithm will work in a real environment. It seems reasonable to verify: i) how it behaves in a dynamic environment, ii) what is a complexity of the proposed RL-based algorithm, iii) how long it takes a decision, iv) what is energy consumption to find the optimum neighbor I recommend to perform some experiments to make the proposal more realistic.
- There are some minor editorial issues, like i) line 186 - please fix repetitions, please correct English.
Author Response
We would like to sincerely thanks the respected reviewer for the time and efforts he/she spent in reviewing our paper. Also, we highly appreciate the valuable comments given by the respected reviewer, which we do believe it enhances the clarity and readability of the revised manuscript. In the following, we have tried our best to address all the comments given by the respected reviewer
Kindly see the attached reply letter

Reviewer 4 Report
The authors propose the use of a RL method to improve the performance of D2D communications for mmWave in B5G/6G technologies. The topic is relevant, but as can be seen below, there are several issues. The manuscript does not reach the standards that would be expected in a Sensors contribution.
First of all, the paper claims that one of the most relevant contributions is the fact that WiFi contextual information is used. However, this is not proved at all, since the authors are using a system-level evaluation. Even if they mention some references, all of them from the same authors, this is not proved in the paper, so it's not clear whether in a real system the use of WiFi information (in 5Ghz, where there might be some interference) would actually be valid to assess the performance over mmWave bands.
Another aspect that should be clarified is the scenario being considered. It seems that the d2d are used so that nodes can access the BS using other as forwarding nodes. Is this right? Does traffic patter have any significance?
On the other hand, the fact that all aspects regarding neighbor discovery, beam steering, etc is not considered at all lead to some doubts regarding the applicability of the proposed scheme. Some discussion should be given at last.
The way energy is considerd in Problem (6) as a constraint would actually not fairly distribute energy, right? You might select one node until it reaches the limit, while there might other potential nodes with more energy that are not selected. Is this a correct interpretation? Shall the scheme work in this way?
It is not clear whether LinCUV and CTS are proposals of this work or they come from references [15] and [16]. This should be clarified. If they have not been proposed in the paper, then theys should be used as benchmark for the EA- solutions, and the results of Fig. 3-5 should be merged with the following ones.
Some more details about the evaluation should be given. It's a system-level numerical evaluation, but since node and obstacle positions are random, did the authors conduct a number of independent experiments per configuration? Would it be possible to corrobate the validity of the results with a complementary approach.
How is the optimal solution selected? Is it by means of exhaustive search? How complex are the proposed algorithms? Could they be used in real scenarios?
The presentation of the manuscript should be improved as well. There were few typos and odd sentences, and a thorough proof-read, preferably with the aid of a professional English speaker is advised. In addition, the authors should use homogenous format for the performance figures. (#5 and #10 look different than the others).
Author Response

(The authors gave the same response as above.)

Round 2
Reviewer 2 Report
The authors have successfully covered the requested comments and changes.
Author Response
We would like to sincerely thanks the respected reviewer for the time and efforts he/she spent in reviewing our paper. Also, we highly appreciate the valuable comments given by the respected reviewer, which we do believe it enhances the clarity and readability of the revised manuscript. We highly appreciate the respected reviewer's comments and thank him too much.

Reviewer 3 Report
Thank you for your explanation and impovements. The paper is a bit better now. However, the main drawback of the paper, that is lack of experiments in a real environment is crucial. Anyway, it could be accepted after language revision.
Author Response
Thanks for your valuable comment. We promise you to do experimental setups in the near future once the related equipments will be available. The whole manuscript was carefully reviewed and corrected against typos and grammatical errors as shown in the following samples.

Reviewer 4 Report
Based on the comments made by all the referees for the first version of the submitted manuscript and the responses made by the authors, there are still several concerns that have not been appropriately dealt with.
1) The contribution of the paper is still not very clear. Two of the reviewers pointed out that the there were a few papers from the same authors (most of them cited in the initial submission) and that it was not straightforward to identify the contribution of this paper. This is not clear yet.
2) One could think that the contribution lies on the fact that WiFi signals are used to estimate the characteristics of the mmWave links. The response on this particular question is not fully satisfactory. The authors claim that this was done in another paper, but the bottomline is that they're using signals at 5 Ghz (most likely without beams) to estimate the characteristics of 60 Ghz links. The features of these two particular channels are so different that having rather theoretical figures to prove that the proposed approach is valid is, in this referee's opinion, not enough.
3) The proposed problem and how it is solved is not very clear. The authors claim the remaining energy at other nodes can fairly distribute the energy, but then the question is: how frequent do they need to solve the problem? The paper says that the "parameters are updated for the next round". When does such next round happen? A more clear discussion of the optimization problem is needed, in any case.
4) The authors say that they are using Montecarlo analysis based on Matlab. This might be ok, but some key information is missing: what is the change between every independent experiment? Is it nodes' locations? How do you model the corresponding channels? Do you consider any randomness there? How many runs did you execute per configuration? It would be worth discussing the statistical tightness of the results.
5) The comment regarding applicability was most likely not questioning the existence of devices with WiFi and mmWave, which might be available in the future, but more about the complexity of finding a solution, and whether this would actully fit with the stringent requirements of 5G/6G communicactions in terms of delay. The authors have briefly discussed the computational complexity of the proposed scheme. The authors claim that d and N are small in the paper, but would this be the case in a real setup? Would the solution scale well?
This referee still believes that such comments are rather serious and that the manuscript should not be accepted.
Author Response
We would like to deeply thank the respected reviewer for the time and efforts he/she spent in reviewing our paper. Also, thanks too much for appreciating our humble work presented in this paper. For the other valuable comments given by the respected reviewer, we tried our best to address all of them, as explained in the attached file.
"Please see the attachment."
